Ecomorphospace occupation of large herbivorous dinosaurs from Late Jurassic through to Late Cretaceous time in North America

Wyenberg-Henzler Taia taiawh626@gmail.com
Department of Biological Sciences, University of Alberta , Edmonton , Alberta , Canada
Meegaskumbura Madhava
Electronic publication date: 2022 Apr 11
Publication date: 2022
Volume: 10
Electronic Location ID: e13174
Received 2021 Nov 26; Accepted 2022 Mar 6
Copyright: ©2022 Wyenberg-Henzler
Copyright year: 2022
Copyright holder: Wyenberg-Henzler
License: This is an open access article distributed under the terms of the Creative Commons Attribution License, which permits unrestricted use, distribution, reproduction and adaptation in any medium and for any purpose provided that it is properly attributed. For attribution, the original author(s), title, publication source (PeerJ) and either DOI or URL of the article must be cited.
License URL: https://creativecommons.org/licenses/by/4.0/

Keywords: Sauropoda, Stegosauria, Ankylosauria, Igaunodontia, Competitive potential, Turnover, Megaherbivore, Early cretaceous, Ecology

Funding: Natural Sciences and Engineering Research Council of Canada Discovery Grant RGPIN-2017-06356 The Dinosaur Research Institute Jurassic Foundation Grant This work was supported by the Natural Sciences and Engineering Research Council of Canada Discovery Grant (RGPIN-2017-06356), the Dinosaur Research Institute (Neoceratopsian Grant), and the Jurassic Foundation Grant (funding awarded in 2019). The funders had no role in study design, data collection and analysis, decision to publish, or preparation of the manuscript.

==============================
Following the Late Jurassic, megaherbivore communities in North America undergo a dramatic turnover in faunal composition: sauropods decline to the point of becoming relatively minor components of ecosystems, stegosaurs become extinct, and hadrosaurids, ceratopsids and ankylosaurs rise in diversity and abundance. Although a variety of causes have been proposed to account for the dramatic decrease in sauropod diversity following the Late Jurassic and could have also been applicable to the disappearance of stegosaurs, the potential for competitive replacement of sauropods by hadrosauroids as an explanation has been previously dismissed due to morphological differences without further investigation. Using twelve ecomorphological correlates of the skull, this study provides a preliminary investigation into ecomorphospace occupation of major megaherbivore clades from the Late Jurassic through to the Late Cretaceous of North America and assess if morphological differences were enough to have potentially facilitated dietary niche partitioning between sauropods and iguanodontians and stegosaurs and ankylosaurs. Overlap in reconstructed ecomorphospace was observed between sauropods (particularly non-diplodocid sauropods) and iguanodontians, as would be expected if morphological differences were not enough to facilitate niche partitioning, contrary to original claims used to dismiss the competitive replacement hypothesis. Overlap was also observed between stegosaurs and ankylosaurs, particularly between Late Cretaceous ankylosaurs. Whether this overlap is reflective competitive replacement or opportunistic occupation of recently vacated niches will require further assessment as sampling of some clades prior to the Late Cretaceous is too poor to make a reliable assessment and several underlying assumptions necessary for competition to occur (e.g., resource limitation) still need investigation. Teasing out the cause(s) of the ‘sauropod decline’ and extinction of stegosaurs in North America following the Late Jurassic will require future research not only into the competitive exclusion hypothesis, but other hypotheses as well with better sampling from Early Cretaceous and Late Jurassic intervals.

Introduction

Large (>1,000 kg) herbivorous dinosaurs (=megaherbivores) dominated terrestrial North American communities for the last ∼98 Myr of the Mesozoic Era both in diversity and abundance (O’Gorman & Hone, 2012; Codron et al., 2012; Codron, Carbone & Clauss, 2013; Brown et al., 2013a). Between the Late Jurassic and Late Cretaceous, a major change in megaherbivore faunal composition is observed: previously dominant sauropods are replaced by ‘duck-billed’ hadrosaurids and ‘horned’ ceratopsids, stegosaurs become extinct and armoured ankylosaurs experience an increase in diversity (Zanno & Makovicky, 2011; Barrett, 2014; Nesbitt et al., 2019; Holtz, 2021). The decline and decrease in sauropod diversity (herein the ‘sauropod decline’) occurred between the Cenomanian and Campanian/Maastrichtian—much later than the complete loss of stegosaurs that appears to have occurred between the Tithonian and Albian (Williamson & Weil, 2008; D’Emic, Wilson & Thompson, 2010; Mannion & Upchurch, 2011; D’Emic & Foreman, 2012).

Several factors have been suggested to explain the Late Jurassic—Late Cretaceous megaherbivore faunal turnover (particularly the ‘sauropod decline’) including changing climate, changing floral composition, changing sea-level, taphonomic biases and competition (D’Emic & Foreman, 2012 and references therein; Barrett, 2014). Many of these non-mutually exclusive hypotheses (e.g., climate, flora, sea-level), are difficult to test using the North American fossil record because the Early-“middle” Cretaceous portion is particularly incomplete. Other hypotheses have been dismissed for various reasons. The taphonomy/sampling hypothesis for example, has been previously dismissed as a complete explanation on the grounds that these patterns (the absence of stegosaurs and reduced diversity/absence of sauropods) are still observed despite increased collection effort (Mannion & Upchurch, 2011; D’Emic & Foreman, 2012). Although some work has more recently investigated the applicability of the taphonomy/sampling hypothesis to the ‘sauropod decline’, the conclusions of such studies are still disagreed upon leaving questions surrounding the validity of the taphonomy/sampling hypothesis still unresolved (e.g.,  Mannion & Upchurch, 2011; D’Emic & Foreman, 2012). The competitive replacement hypothesis was also dismissed when it was originally proposed by Lucas & Hunt (1989) because the morphologies of sauropods and ornithischians were thought to have been too different for competition to have been possible. However, the presence of morphological differences does not mean two groups were unable to compete for the same resources: organisms can differ in morphology and yet still accomplish the same task. A lack of competitive potential due to morphological dissimilarity should be demonstrated on an ecomorphological basis- especially when spatiotemporal overlap is difficult to reject. For example, despite mammals, birds and lizards possessing widely different body plans, species consuming tough vegetation (e.g., seeds) will tend to possess short skulls and deeps jaws as these features are useful for generating and resisting high bite forces necessary for the consumption of mechanically resistant plant vegetation (Greaves, 1974; Janis, 1990; Janis, 1995; Greaves & Thomason, 1995; Spencer, 1995; Mendoza, Janis & Palmqvist, 2002; Metzger & Herrel, 2005; Herrel et al., 2005; Herrel et al., 2006; Mitchell et al., 2018). Conversely, different species of finches will not exhibit dietary overlap despite possessing largely similar morphologies due to slight differences in beak shape (e.g., length, depth, curvature) (Bowman, 1961; Herrel et al., 2005). Thus, demonstrating a lack of dietary overlap requires consideration of traits that are reflective of basic form-function relationships rather than comparisons between gross morphologies. Further, some researchers have more recently suggested that the earlier dismissal of the competitive replacement hypothesis was premature given the masticatory capabilities of hadrosauroids would have permitted a wide niche breadth and recognition of niche shifts through ontogeny (D’Emic & Foreman, 2012; Erickson & Zelenitsky, 2014; Wyenberg-Henzler, Patterson & Mallon, 2022). Yet, work assessing if morphological dissimilarity was great enough between hadrosauroids and sauropods to have precluded competition remains to be conducted. In addition, how these and other major megaherbivorous clades are positioned relative to one another in an ecomorphospace reconstructed from variables taken from the entire skull and considering taxa exclusively from North America through time has yet to be investigated. Using linear morphometrics and snout shape proxies to reconstruct megaherbivorous dinosaur ecomorphospace, this study aims to provide a preliminary investigation into ecomorphospace occupation of major megaherbivore clades from the Late Jurassic through to the end of the Late Cretaceous and assess if there were ecomorphological differences between major clades of megaherbivorous dinosaurs that may have potentially resulted in resource partitioning rather than competition assuming that other conditions, such as resource limitation, necessary for competitive antagonism to occur are met. Should ecomorphological differences be established this would suggest competitive replacement was not a plausible explanation for the ‘sauropod decline’ and extinction of stegosaurs in North America as resources were more likely to have been partitioned between clades as was originally claimed by Lucas & Hunt (1989).

The competitive replacement hypothesis of Lucas & Hunt (1989) specifically proposes that hadrosauroids outcompeted and eventually replaced sauropods. For this to have occurred, hadrosauroids or their ancestors would have needed to arrive in North America prior to the Cenomanian ‘sauropod decline’ (Williamson & Weil, 2008; D’Emic, Wilson & Thompson, 2010; Mannion & Upchurch, 2011; D’Emic & Foreman, 2012). Fossil evidence indicates hadrosauroids were present in North America during the early Cenomanian (Sues & Averianov, 2009; Prieto-Márquez, 2010; McDonald, Wolfe & Kirkland, 2010; McDonald et al., 2012; Prieto-Márquez, Erickson & Ebersole, 2016) which is arguably in line with this prediction. More basal iguanodontians (e.g., Iguanocolossus) arrive in North America no later than the Barremian (McDonald et al., 2012; Xu et al., 2018) increasing the likelihood that hadrosauroids, and/or even their ancestors, the non-hadrosauroid iguanodontians, may have played a role in the decline in sauropod diversity.

Competitive replacement may also explain the Late Jurassic/Early Cretaceous stegosaur extinction in North America. Stegosaurs, known from the Late Jurassic Morrison Formation, disappear from Early Cretaceous deposits. At around this same time ankylosaurs experience an increase in diversity (Romano, 2021). Stegosaurs and ankylosaurs are also interpreted as low browsers (∼1 m) (Foster, 2003; Mallon et al., 2013). However further investigation is needed to assess the potential for competition between stegosaurs and ankylosaurs as niche partitioning has been proposed to have facilitated megaherbivore coexistence both in the fauna of the Late Jurassic Morrison Formation and in Late Cretaceous ecosystems (e.g.,  Foster, 2003; Whitlock, 2011; Mallon, 2019). Investigating the distribution of major megaherbivore clades within ecomorphospace will not only help assess if it was even conceivably possible for sauropods and stegosaurs to have been competitively replaced by iguanodontians and ankylosaurs respectively, but also provide information on how the structuring of megaherbivore ecosystems may have changed through time within North America.

Materials & Methods

All analyses were conducted using the programming language R v. 4.0.2 (R Core Team, 2019) (available in Code S1 and S2). All packages used for these analyses are available in File S1 and listed at the top of Code S1 and S2.

Brief comment on terminology and ecological theory

Many palaeontological studies investigate ecomorphospace occupation of an assemblage to approximate niche relationships (Van Valkenburgh, 1994; Stroik, 2014). This assumes that the relationship between form and function, imperfect as it may be (Lauder, 1995), is strong enough to reflect the ecological niche of the organisms under study. Within this context, competitive potential (i.e., that groups could have competed) is established when niches of two or more groups overlap (Stroik, 2014 and references therein). In the present study, the term niche is used to refer to the dietary ecology of a group or groups and ecomorphology to refer to aspects of the morphology related to dietary ecology. To refer to the broader concepts associated with dietary and non-dietary niches and dietary and non-dietary ecomorphology, I will use the phrases “general niche” and “general ecomorphology”.

Within the context of general niche theory, in order for competitive exclusion to be possible several conditions must be met. This includes overlap not only in dietary requirements but also in time and space as well as resource limitation (Stroik, 2014 and references therein). For the purposes of the present study, I have assumed that these and some other important conditions (see ‘Assumptions’) necessary for competition to occur are met as Lucas & Hunt (1989) had originally dismissed the competitive exclusion hypothesis on the grounds of morphological dissimilarity and by extension lack of dietary overlap. It should also be made clear that even if overlap is observed between clades in ecomorphospace, that this does not mean competition was likely to have occurred and been responsible for the ‘sauropod decline’ and North American stegosaur extinction. Rather, this overlap would be indicative that the overlapping clades could have potentially competed on the basis of ecomorphological similarity but the validity of the competitive exclusion hypothesis would need further testing to assess if overlap along other non-dietary niche dimensions was also present.

Assumptions

To begin to assess if morphological disparity alone was enough to have precluded competition between megaherbivore clades, especially sauropods and iguanodontians (sensu Lucas & Hunt, 1989), several simplifying assumptions are made but should be tested in future assessments of the competitive replacement hypothesis as a whole. The first involves resource availability. Although multiple studies on Mesozoic North American megaherbivore communities have established that resources were likely limiting and/or that niche partitioning was present (e.g.,  Foster, 2003; Mallon & Anderson, 2013; Mallon, 2019), resource limitation should be established for the entirety of the Late Jurassic through to the end of the Cretaceous.

Overlap in time and space is also assumed between taxa in the coarse temporal bins used (discussed below). The taxa included here are sampled from various localities across Alberta, Saskatchewan, Montana, North Dakota, Wyoming, Texas, Utah, Kansas and New Mexico that span from the Upper Jurassic Morrison Formation to the Upper Cretaceous Hell Creek Formation encompassing around ∼90 million years in duration (from 156 to 66 Ma) (Geological Society of America, 2018; Maidment & Muxworthy, 2019) (Table 1). Given very few specimens from the clades of interest from the Early Cretaceous of North America have been recovered and published on in the literature, and the extinction of North American stegosaurs prior to the Early Cretaceous, it is only possible to conduct a preliminary investigation into ecomorphospace at this time. For simplicity, I have split the ecomorphological dataset (see next section for details) into three main blocks of time: Late Jurassic (156–147 Ma; Kimmeridgian-Tithonian; Maidment & Muxworthy, 2019), Early Cretaceous (145.0–100.5 Ma; Barriasian-Albian; Geological Society of America, 2018), and Late Cretaceous (100.5–66 Ma; Cenomanian-Maastrichtian; Geological Society of America, 2018). Although some time bins are easier to split into smaller intervals while maintaining relatively larger (n > 5) sample sizes for each clade (e.g., Campanian Dinosaur Park Formation), other time intervals such as the Early Cretaceous, have sample sizes for some clades that are too small (n < 3) to reasonably separate into smaller time slices. Further, stegosaurs and most sauropods disappear after the Early Cretaceous and would not be represented in subsequent time periods preventing any comparison in ecomorphology between potential replacement clades (e.g., iguanodontians). There is also the added difficulty of finding suitable material for the groups considered that have been collected from the same formation, or even more ideally, the same locality.

Table 1 Information for specimens included in the principal component analyses.

Clade	Genus	Sample size	Ontogenetic stage(s)	Formation(s)	
Late Jurassic	
Stegosauria	Stegosaurus	2	Adult	Morrison	
Hesperosaurus	4	Adult	Morrison	
Ankylosauria	Gargoyleosaurus	1	Adult	Morrison	
Sauropoda (diplodocid)	Apatosaurus	2	Adult (1) subadult (1)	Morrison	
	Diplodocus	2	Adult (1) subadult (1)	Morrison	
	Galeamopus	1	Adult	Morrison	
Kaatedocus	1	Adult	Morrison	
Sauropoda (non-diplodocid)	Camarasaurus	6	Adult (5) subadult (1)	Morrison	
Early Cretaceous	
Ankylosauria	Pawpawsaurus	1	Adult	Paw Paw	
	Gastonia	1	Adult	Cedar Mountain	
	Silvisaurus*	1	Adult	Dakota	
	Tatankacephalus	1	Adult	Cloverly	
	Crichtonpelta*	1	Adult	Sunjiawan (Ch)	
	Gobisaurus	1	Adult	Ulanhushao (Mo)	
Shamosaurus	1	Adult	Zuunbayan (Mo)	
Sauropoda (non-diplodocid)	Abydosaurus	1	Adult	Cedar Mountain	
	Euhelopus	1	Adult	Mengyin (Br)	
Tapuisaurus	1	Adult	Quirico (Br)	
Iguanodontia (basal)	Tenontosaurus	3	Adult (2) subadult (1)	Cloverly/ Antlers	
	Jinzhousaurus	1	Adult	Yixian (Ch)	
	Xuwulong	1	Adult	Xinminpu (Ch)	
	Chyorodon	1	Adult	Khuren Dukh (Mo)	
	Altirhinus	1	Adult	Khuren Dukh (Mo)	
Proa	1	Adult	Escucha (Sp)	
Iguanodontia (Hadrosauroidea)	Equijubus	1	Adult	Middle Grey Unit of Xinminbao Group (Ch)	
Eolambia	1	Adult	Cedar Mountain	
Late Cretaceous	
Ankylosauria	‘Zhongyuansaurus’*	1	Adult	Ulansuhai (Ch)	
	Talarurus*	2	Adult	Bayanshiree (Mo)	
	Tsagantegia*	1	Adult	Bayanshiree (Mo)	
	Pinacosaurus*	1	Adult	Djadokhta (Mo)	
	Saichania	4	Adult	Barun Goyot (Mo)	
	Tarchia	2	Adult	Nemegt (Mo)	
	Euoplocephalus	9	Adult	Dinosaur Park	
	Edmontonia	2	Adult	Dinosaur Park	
	Panoplosaurus	8	Adult	Dinosaur Park	
	Zuul	1	Adult	Judith River	
	Ziapelta	1	Adult	Kirtland	
Ankylosaurus	2	Adult	Scollard	
Ceratopsidae	“pachyrhinosaur”	1	Adult	Dinosaur Park	
	Centrosaurus	12	Adult (11) subadult (1)	Dinosaur Park	
	Styracosaurus	1	Adult	Dinosaur Park	
	Chasmosaurus	11	Adult (9) subadult (2)	Dinosaur Park	
	Pachyrhinosaurus	1	Adult	Horseshoe Canyon	
	Anchiceratops	3	Adult	Horseshoe Canyon	
Triceratops	14	Adult (10) subadult (4)	Hell Creek/ Lance	
Sauropoda (non-diplodocid)	Sarmientosaurus*	1	Adult	Bajo Barreal (Ar)	
Nemegtosaurus	1	Adult	Nemegt (Mo)	
Iguanodontia (Hadrosauroidea)	Plesiohadros*	1	Adult	Djadokhta (Mo)	
Protohadros	1	Adult	Woodbine	
Iguanodontia (Hadrosauridae)	Eotrachodon	1	Adult	Eutaw	
	Brachylophosaurus	3	Adult	Judith River/ Oldman	
	Gryposaurus	6	Adult (4) subadult (2)	Dinosaur Park/ Kaiparowits	
	Prosaurolophus	7	Adult (5) subadult (2)	Dinosaur Park	
	Corythosaurus	19	Adult (15) subadult (4)	Dinosaur Park	
	Lambeosaurus	15	Adult (13) subadult (2)	Dinosaur Park	
	Parasaurolophus	2	Adult (1) subadult (1)	Dinosaur Park	
	Hypacrosaurus	6	Adult (3) subadult (3)	Horseshoe Canyon/ Two Medicine	
	Maiasaura	4	Adult (3) subadult (1)	Two Medicine	
Edmontosaurus	24	Adult (18) subadult (6)	Horseshoe Canyon/ Hell Creek/ Frenchman	
Notes.

Listed are formations that specimens included in the study were collected from. Sample size reflects the total sample size and of various ontogenetic stages are represented in the sample the sample size for each stage is given in brackets next to the corresponding stage in the ontogenetic stage(s) column. All taxa are from North American formations unless otherwise noted.

Ar taxa from Argentina

Br taxa from Brazil

Ch taxa from China

Mo taxa from Mongolia

Sp taxa from Spain

* Taxa from the early Late Cretaceous or late Early Cretaceous to early Late Cretaceous.

Any investigation into potential competitive interactions among herbivores must also consider potential differences in feeding heights, as such differences have been shown to facilitate resource partitioning (Lamprey, 1963; Bell, 1971; Leuthold, 1978; McNaughton & Georgiadis, 1986; Coe et al., 1987; Foster, 2003; Mallon et al., 2013; Barrett, 2014). Stegosaurs, ankylosaurs and ceratopsids have been previously suggested to have fed on vegetation at or below 1 m in a quadrupedal posture (but see Mallison, 2010 for alternative feeding poses in stegosaurs), whereas hadrosaurids (and presumably other similar-sized iguanodontians) are reconstructed as having fed up to 4 m and Diplodocus, Apatosaurus, Camarasaurus and Brachiosaurus are variably reconstructed with feeding heights below 2.9 or 3.4 m, below 3.4 or 5.3 m, between 3 and 5.7 m, and between 5 and 9.4 m, respectively (Foster, 2003; Mallon et al., 2013; Barrett, 2014).

Ecomorphospace reconstruction and analysis

general procedure and data collection

Twelve cranial measurements known to correlate with feeding behaviour, feeding height, plant mechanical properties and other aspects of feeding ecology in modern animals (e.g., Janis, 1990; Janis, 1995; Spencer, 1995; Mendoza, Janis & Palmqvist, 2002; Mitchell et al., 2018) and have been previously used in other dinosaur ecology studies (e.g., Whitlock, 2011; Mallon & Anderson, 2013; Mallon & Anderson, 2015; MacLaren et al., 2017; Osi et al., 2017; Mallon, 2019; Wyenberg-Henzler, 2020; Wyenberg-Henzler, Patterson & Mallon, 2021; Wyenberg-Henzler, Patterson & Mallon, 2022) (see Fig. 1 and Table S1 for details) were used assess niche overlap. These measurements were collected from the literature and/or by travelling to museums to measure in person (AMNH, CMN, GPDM, MOR, ROM, TMP, UALVP, USNM) (Table S1).

Figure 1 Measurements used in ecomorphological analysis in sauropods.

Lateral view of measurements taken from Diplodocus (A) and Camarasaurus (B). (C) Dorsal view of Diplodocus snout with lines used to obtain SSI (shaded stippled region) using the modified method of Dompierre & Churcher (1996). (D) Posterior view of measurements taken from the skull for Diplodocus. Measurements taken from iguanodontians, ceratopsids, ankylosaurs and stegosaurs are based on those from Mallon & Anderson (2013) and Wyenberg-Henzler (2020). Description of measurements used are available in Table S1. Diplodocus outline produced from photos of SMA 0011 in Tschopp & Mateus (2017) and Camarasaurus outline produced from images of CMNH 11338 in Button, Rayfield & Barrett (2016). Abbreviations: a1, a2, c, bounding lines for snout shape used to obtain SSI; as-mq, anterior snout to middle quadrate distance; b, line used to define hypotenuse of the triangle used to obtain SSI; cp-jj, coronoid process to jaw joint distance; cs-mq, cropping surface to middle quadrate distance; mt-mq, mesial tooth row to middle quadrate distance; dh, dentary height; dt-mq, distal tooth row to middle quadrate distance; oh, occiput height; ppb, paroccipital process breadth; qb, quadrate breadth; sk, skull height; sw, snout width (after Mallon & Anderson, 2013; Wyenberg-Henzler, 2020).

Snout shape index was taken following the modified methodology of Dompierre & Churcher (1996) used in Wyenberg-Henzler (2020) and (Wyenberg-Henzler, Patterson & Mallon, 2021; Wyenberg-Henzler, Patterson & Mallon, 2022). This snout shape proxy uses photos of the snout taken in dorsal or ventral view to distinguish between wide, square shaped snouts and narrow, pointed snouts. Outlines taken of the snout are overlaid by a right-angle triangle with two sides bordering the anterior-most extension and widest part of the snout with a hypotenuse at an angle of 26° from the vertical (Fig. 1). The ratio of the area of the triangle anterior the outline of the snout and the area of the triangle itself yields the snout shape index which reflects the relative broadness of the snout—a feature known to reflect feeding selectivity in modern ungulates (e.g., Spencer, 1995; Dompierre & Churcher, 1996).

To linearize relationships between variables, numerical data were log-transformed prior to subsequent analyses. Due to the incomplete nature of most specimens and the nature of the analyses, only specimens for which ≥50% of these measurements could be taken were included, subset by family, and subjected to Bayesian principal components analysis (PCA) to produce a dataset without missing values (Hammer & Harper, 2006; Brown, Arbour & Jackson, 2012) (Code S1, File S1). Bayesian PCA is complex method wherein missing values are iteratively calculated from known values using PCA expectation maximization combined with a Bayesian model (Brown, Arbour & Jackson, 2012 and references therein). The completed dataset was then subjected to a separate PCA using a correlation matrix for principal component axis construction to reconstruct a theoretical Late Jurassic—Late Cretaceous ecomorphospace.

Convex hulls were drawn around iguanodontians (basal Iguanodontia, Hadrosauroidea, Hadrosauridae), diplodocids, non-diplodocid sauropods (Camarasauridae, Brachiosauridae), stegosaurs, ceratopsids and ankylosaurs as these were the major subgroupings of interest. The hulls were drawn around all data points for each group, ignoring time differences to see how individuals from different time periods plotted relative to their respective groups as a whole (see ‘Assumptions’). It is also possible to make comparisons between clades that would have had coeval representatives but lack enough representative material to be included in the analysis (e.g., Late Jurassic sauropods and Early Cretaceous iguanodontians).

To determine if groups from the Late Jurassic/Early Cretaceous showed significant overlap, a non-parametric analysis of variance (NPMANOVA) was conducted on a rarified subset (n = 6 per group) of PC scores for all axes individually accounting for >5% of the total variation (Code S1, File S1). This process was repeated over 1,000 iterations and the resulting p-values were combined using the harmonic mean p method, which is designed to combine multiple p-values from non-independent tests into a single value (Wilson, 2019). Two specific tests were conducted, one comparing stegosaur PC scores to scores from other clades (e.g., ceratopsids, ankylosaurs) showing overlap with the stegosaurs both within reconstructed ecomorphospace and in reconstructed feeding height (see ‘Assumtions’), and another comparing sauropods to iguanodontians (assuming overlap is observed). Conducting follow-up analyses in this manner increases statistical power by limiting the number of pairwise comparisons being made and therefore reducing the degree of correction needed to control for familywise error rate (Hammer & Harper, 2006). If the omnibus test comparing all sauropods to all iguanodontians was significant (α = 0.05), follow-up pairwise comparisons were conducted between diplodocids, non-diplodocid sauropods and iguanodontians using rarified datasets over 1,000 iterations, combined using the harmonic mean p method, and then corrected for multiple comparisons with Holm-correction (Holm, 1978). Pairwise comparisons were not conducted between all taxa as differences in feeding height (see ‘Assumptions’) would have likely facilitated niche partitioning between certain clades and increasing the number of pairwise comparisons reduces statistical power.

Analytical specifics and inclusion of non-North American taxa

The methods described in the previous section (2.3.1 general procedure and data collection) were conducted on two different datasets. The first dataset contains ecomorphological proxies taken from North American taxa exclusively (Table S2). The second dataset contains measurements of ecomorphological proxies for these same North American taxa (Table S2), as well as measurements of non-North American taxa from the Early Cretaceous and Late Cretaceous (Table S3).

The use of non-North American taxa in this second dataset to approximate what North American taxa from these poorly sampled time periods (see ‘Assumptions’) is justified given at least one interchange event between North America and adjacent landmasses is proposed to have occurred between the Early Cretaceous—early Late Cretaceous. Titanosaur sauropods and iguanodontians in particular have featured in these discussions regarding faunal interchange between North America from Asia, Europe, and/or South America around the time of the ‘sauropod decline’ observed in North America (e.g., McDonald, Wolfe & Kirkland, 2010; Zanno & Makovicky, 2011; Buffetaut & Suteethorn, 2011; D’Emic & Foreman, 2012; Tykoski & Fiorillo, 2017; Xing, Mallon & Currie, 2017) (but see also Mannion & Upchurch, 2011). Because of the uncertainty associated with the number and landmasses involved in this/these origin event(s), any representatives from the various clades which are poorly represented during the Early Cretaceous—early Late Cretaceous were included in the second dataset analyzed (and are indicated in the list of taxa in Table 1). Presumably, taxa from these regions at this time would share similar ecomorphologies with taxa that were present in North America during this time, and could thus be considered reasonable proxies.

Phylogenetic considerations

Phylogenetic similarity between ankylosaurs, stegosaurs, ceratopsids and iguanodontians may also influence the results as more closely related groups will be more similar to each other due to shorter divergence times (Pagel, 2002; Symonds & Blomberg, 2014). Although phylogenetic similarity is certainly important in the assembly of animal bauplans, I have opted to not apply phylogenetic corrections to the data presented here for several reasons. First, there is some evidence to suggest that certain traits evolve in animals consuming the same resource, even in distantly related clades. For example, herbivores consuming mechanically resistant foodstuffs will tend to evolve short skulls and deep jaws regardless of phylogenetic relatedness (Bowman, 1961; Anapol & Lee, 1994; Nogueira et al., 2005; Mallon & Anderson, 2013). This is because short skulls position the bite point closer to the adductor musculature increasing mechanical advantage, and subsequently bite force, whereas deeper jaws are well suited for resisting high resistive forces generated when hard materials are bitten into (Greaves, 1974; Janis, 1990; Janis, 1995; Greaves & Thomason, 1995; Spencer, 1995; Mendoza, Janis & Palmqvist, 2002; Metzger & Herrel, 2005; Herrel et al., 2006; Mitchell et al., 2018). Selecting skull measurements shown to be applicable across a wide range of taxa (including Mammalia and Reptilia) and are informed by such form-function relationships helps to mediate the influence of phylogeny on ecological inferences. Second, application of a phylogenetic correction to the data, such as a phylogenetic principal components analysis (pPCA), does not necessarily correct phylogenetic signal and third, applying a phylogenetic correction can introduce new issues. For example, pPCA does not entirely remove phylogenetic signal from the dataset as the ‘corrected’ values produced are still plotted within the original shape space (Polly et al., 2013). Additionally, pPCA produces axes that are not independent from each other adding computational issues to subsequent statistical analyses (e.g., NPMANOVA). For these reasons, and for simplicity, I chosen to employ traditional analytical techniques that do not apply phylogenetic correction and so the results presented here should be interpreted accordingly.

Body size distribution through time

Body size is an important trait to consider in ecological research as it has been tied to numerous other variables such as species abundance, metabolic rate, energy requirements, reproductive mode and resource use (Colinvaux, 1979; Peters & Wassenberg, 1983; LaBarbera, 1989; Griffiths, 1992; Blackburn & Gaston, 1994; Brown, 1995; Gaston & Blackburn, 2000; Ernest, 2005; White et al., 2007). Several studies have noted that (at least some) dinosaurian ecosystems appear to exhibit species-richness body size distributions that bias towards larger species, contrary to those observed in mammalian communities. Whether this signal is more a reflection of differences in community structure resulting from differences in life history (e.g., Codron et al., 2012; Codron, Carbone & Clauss, 2013; Schroeder, Lyons & Smith, 2021; Holtz, 2021) or more of an artefact of taphonomic biases (e.g., Brown et al., 2013a; Brown et al., 2013b), is still debated and beyond the scope of the present study. However, because body size is such an important ecological factor, even preliminary consideration into potential differences in body size-species richness distributions between dinosaur assemblages through time is necessary to further elucidate ecological interactions leading up to, during, and after the ‘sauropod decline’ in North America.

Although other analyses such as an “ecological structure analysis” (see Noto & Grossman, 2010) or browse profile analysis (Coe et al., 1987; Wyenberg-Henzler, Patterson & Mallon, 2021) would provide more information than simple body size species-richness distributions would, given uncertainties associated with feeding heights and the incompleteness of taxonomic assemblages from key time intervals (see ‘Assumptions’), it is presently not feasible to conduct such analysis—although it would be prudent to do so in the future. Body mass data for ornithischians and sauropods from select Late Jurassic—Late Cretaceous formations were amassed from the literature (see Tables S3 and S4). Pre-Late Cretaceous North American formations were selected if there were multiple (n > 5) taxa known with available body mass data. To make comparisons between time periods easier, only a handful of ‘representative’ formations from Late Cretaceous North America were selected because they have been well-sampled (e.g., Dinosaur Park, Two Medicine, Horseshoe Canyon, Hell Creek formations), or are some of the few Late Cretaceous North American formations not from the north/north-western portion of the continent that meet the minimum number of taxa threshold (e.g., Kaiparowits, Aguja, Kirtland, Ojo Alamo formations). Several of the better sampled non-North American formations from the Middle Jurassic—Late Cretaceous (e.g., Shaximiao, Yixian, Nemegt formations) were also included for comparison as assemblages from other continents do not have appeared to undergo a ‘sauropod decline’. Body masses were log10-transformed, with the type of skewness evaluated and kernel density distribution overlain on a histogram. Skewness type was determined according to the categories provided in Bulmer (1979) where skewness (s), is considered strong when —s— ≥ 1, moderate when 0.5 ≤ —s— < 1, and weak when —s— < 0.5.

Results

Broad relationships between North American clades

The first four PCs account for 89.5% of the variation within the North American dataset (PC1 = 58.5%, PC2 = 15.1%, PC3 = 9.4%, PC4 = 6.5%) (Fig. 2, Fig. S1). All variables considered weight positively with increasing values along PC1. Along PC2 distal tooth row length and snout width increases while paroccipital process breadth decreases. Higher values on PC2 also reflect narrower, more pointed snouts (i.e., lower SSI values). Ignoring time intervals, most taxa overlap on PC1 and PC2, with the exception of ceratopsids which do not overlap with stegosaurs and diplodocid sauropods (Fig. 2, Fig. S2). Along PC3 across all time bins most clades show overlap except for ankylosaurs which show some separation from other taxa (mainly stegosaurs), reflecting wide snouts and posterior skull widths and short occiputs of the former. Stegosaurs, diplodocids and to some extent non-diplodocid sauropods show separation from ankylosaurs and iguanodontians along PC4 reflecting relatively long distal tooth row lengths, narrower and less square snouts, and shorter occiputs observed in the former set of clades (Figs. S1 and S2). Ceratopsids plot more centrally along the axis and overlap with all taxa along PC4.

Figure 2 Principal component plots for the first three axes representing reconstructed ecomorphospace during the Late Cretaceous (A), Early Cretaceous (B) and Late Jurassic (C) for North American taxa.

Convex hulls are drawn around specimens from major clades from North America across all time periods. Original image credits for silhouettes: modified from Jagged Fang Designs (Diplodocidae); M.P. Taylor (high-browsing sauropod); Jagged Fang Designs (Iguanodontia/Hadrosauroidea, Ceratopsidae); B. McFeeters and T.M Keesey (Ankylosauria); modified from T. Dixon (Stegosauridae). Measurement and descriptions of raw data are available in Fig. 1 and Tables S1 and S2.

Comparison of North American clade distribution through time

From the Late Jurassic to the Late Cretaceous, North American ankylosaurs exhibit a shift towards the ecomorphospace occupied by Late Jurassic stegosaurs (potentially due to increased sampling of ankylosaurs through time) as reflected by increases in overall skull dimensions along PC1 and increases in distal tooth row length and snout width, decreases in paroccipital process breadth, and widening and squaring of the snout along PC2. However, ankylosaurs continue to remain separate from stegosaurs along PCs 3 and 4 due to differences in posterior skull width, snout width and occiput height.

North American iguanodontians expand their occupied ecomorphospace during the Late Cretaceous primarily along PCs 1 and 3 reflecting increases in overall skull dimensions and shortening occiput height, decreases in the distance between the coronoid process and jaw joint and distance between quadrates. The original ecospace occupied by more basal representatives are later occupied by immature hadrosaurids in the Late Cretaceous. Early Cretaceous iguanodontians almost completely overlap with time-averaged non-diplodocid sauropod ecospace along the first three PC axes and show minor overlap along PC4. Overlap along PCs 1 through 3 is even more pronounced between sauropods across all time bins and Late Cretaceous iguanodontians (especially Late Cretaceous hadrosauroids and immature hadrosaurids).

Figure 3 Principal component plots for the first three axes representing reconstructed ecomorphospace during the Late Cretaceous (A), Early Cretaceous (B) and Late Jurassic (C) for North American and non-North American taxa.

Convex hulls are drawn around specimens from major clades across all time periods. Abbreviations denoting location of non-North American taxa: A, Argentina; B, Brazil; C, China; M, Mongolia; S, Spain. Original image credits for silhouettes after Fig. 2. Measurement descriptions and raw data are available in Fig. 1 and Tables S1, S2 and S3.

influence of the inclusion of non-North American taxa on PCA results

The first four PCs for the analysis of the second dataset account for 89.9% of the variation within the dataset (PC1 = 59.7%, PC2 = 14.6%, PC3 = 9.4%, PC4 = 6.2%) (Fig. 3, Fig. S3). Variable loadings along PC axes remain unchanged with the inclusion of non-North American taxa in the analysis (Fig. S4). Relationships between clades remain largely similar to relationships observed between clades for the North American PCA. Non-North American ankylosaurs and iguanodontians plot between or very close to points representing North American members of the same clade (Figs. 2 and 3, Figs. S1 and S3). The main difference between the North American PCA (Fig. 2, Fig. S1) and the all-inclusive PCA (Fig. 3, Fig. S3) is the ecospace occupied by non-diplodocid sauropods. Two Brazilian specimens and one Mongolian specimen expand the occupied ecospace of non-diplodocid sauropods along PCs 2, 3 and 4. This expansion of occupied ecospace of non-diplodocid sauropods along PCs 2, 3 and 4 increase the relative amount of overlap with iguanodontians.

Results of NPMANOVA

The degrees of freedom, test-statistic and raw-p-values for each test and comparisons conducted for the North American dataset are available in Tables S6–S11 and for the combined non-North American and North American dataset are available Supplementary Tables S12–S17. Comparisons between stegosaurs and ankylosaurs were significant (harmonic mean p = 0.003) even when Late Cretaceous ankylosaurs were excluded (harmonic mean p = 0.002) regardless of which dataset was used. Omnibus tests comparing iguanodontians to sauropods were also significant (harmonic mean p = 0.003) even when adult Late Cretaceous hadrosaurids were excluded (harmonic mean p = 0.002) to assess if their absolutely larger size was impacting the amount of overlap between groups, particularly along PC1 regardless of which dataset was used. Follow-up pairwise-comparisons between North American diplodocids, non-diplodocid sauropods and iguanodontians indicated significant differences among all groups regardless of whether or not adult Late Cretaceous hadrosaurids were included as would be expected for taxa occupying distinct areas of morphospace (Tables 2 and 3). These pairwise comparisons were also significant for the second dataset containing both North American and non-North American taxa (Tables 4 and 5).

Table 2 Harmonic mean p-values for pairwise NPMANOVAs conducted on PC scores for North American iguanodontians and sauropods from the Late Jurassic, Early Cretaceous and Late Cretaceous.

The degrees of freedom, test-statistic and raw p-value for each iteration and each comparison can be found in Tables S10.

Comparison	Uncorrected	Holm-corrected	
Iguanodontians vs diplodocids	0.003	0.008	
Iguanodontians vs non-diplodocids	0.003	0.008	
Diplodocids vs non-diplodocids	0.003	0.008	

Table 3 Harmonic mean p-values for pairwise NPMANOVAs conducted on PC scores for North American non-hadrosaurid iguanodontians and sauropods from the Late Jurassic and Early Cretaceous and immature hadrosaurids from the Late Cretaceous.

The degrees of freedom, test-statistic and raw p-value for each iteration and each comparison can be found in Table S11.

Comparison	Uncorrected	Holm-corrected	
Iguanodontians vs diplodocids	0.003	0.008	
Iguanodontians vs non-diplodocids	0.006	0.008	
Diplodocids vs non-diplodocids	0.003	0.008	

Table 4 Harmonic mean p-values for pairwise NPMANOVAs conducted on PC scores for North American non-hadrosaurid iguanodontians and sauropods from the Late Jurassic and Early Cretaceous and immature hadrosaurids from the Late Cretaceous.

The degrees of freedom, test-statistic and raw p-value for each iteration and each comparison can be found in Table S16.

Comparison	Uncorrected	Holm-corrected	
Iguanodontians vs diplodocids	0.003	0.008	
Iguanodontians vs non-diplodocids	0.004	0.008	
Diplodocids vs non-diplodocids	0.003	0.008	

Table 5 Harmonic mean p-values for pairwise NPMANOVAs conducted on PC scores for combined dataset of North American and non-North American non-hadrosaurid iguanodontians and sauropods from the Late Jurassic and Early Cretaceous and immature hadrosaurids from the.

The degrees of freedom, test-statistic and raw p-value for each iteration and each comparison can be found in Table S17.

Comparison	Uncorrected	Holm-corrected	
Iguanodontians vs diplodocids	0.003	0.008	
Iguanodontians vs non-diplodocids	0.005	0.008	
Diplodocids vs non-diplodocids	0.003	0.008	

Body size distributions

No clear relationship between the shape of the body size species-richness distribution and presence/absence of sauropods, time or location is readily apparent from the body size distributions plots (Fig. 4). Through time, formations show variation in skewness ranging from weakly to strongly skewed. Both sauropod-bearing and non-sauropod-bearing formations show skewness ranging from weak to strong. Sauropod-bearing formations have a greater maximum body size than non-sauropod-bearing formations.

Figure 4 Body size distributions for select formations from the Late Jurassic through to the Late Cretaceous.

Sauropod silhouette denotes formations from which sauropods have been recovered. Note: the Tendaguru formation spans across the Jurassic-Cretaceous boundary. Abbreviations: fm, formation; s, skewness. Raw data and information available in Tables S4 and S5. Original image credits for silhouette: M.P. Taylor.

Discussion

Inclusion of non-North American taxa in analyses

The results of analyses conducted for the dataset of North American taxa and North American + non-North American taxa indicate that the inclusion of non-North American taxa have very little influence on the ecological inferences and conclusions drawn from the data. Therefore, all interpretations given below can be considered applicable to both sets of analyses unless otherwise noted.

Plausibility for competition from a dietary perspective

Results of the NPMANOVAs suggest that Late Jurassic taxa were not competitively excluded by the ancestors of Late Cretaceous ceratopsids, ornithopods and ankylosaurs. These results are perhaps unsurprising as the competitive replacement hypothesis was originally dismissed on the grounds of morphological dissimilarity (Lucas & Hunt, 1989) meaning significant differences in overall shape should be expected. It is also possible that the medians of compared clades do significantly differ despite there being meaningful overlap present as an NPMANOVA only compares the medians (Hammer & Harper, 2006). Making an accurate assessment of competitive potential may therefore require a more nuanced approach wherein morphometric variables weighting on PC axes are interpreted within an ecological context rather than just simply in terms of separation versus overlap.

Stegosaurs vs ankylosaurs

The general placement of stegosaurs by ankylosaurs within ecomorphospace through time combined with observed overlap along PC1, PC2, and PC4, which account for ∼80% of the total variation in the dataset (Figs. 2 and 3, Figs. S2 and S4) and similarities in reconstructed feeding height provided in other studies (Foster, 2003; Mallon & Anderson, 2013), would suggest there was competitive potential, at least from a dietary perspective between these groups. However, it is also possible that ankylosaurs opportunistically invaded stegosaur morphospace after stegosaurs had already been extirpated by other factors (e.g., changing climate). Although the lack of overlap between stegosaurs and ankylosaurs during the Late Jurassic would lend greater support for the opportunistic occupation of niche space following prior extirpation, it is also possible that the lack of overlap between groups is an artifact reflecting small sample sizes, especially for ankylosaurs (n = 1), at this time rather than being reflective of true niche separation.

Sauropods vs iguanodontians

Overlap in ecomorphospace observed between sauropods (particularly non-diplodocid sauropods) and basal iguanodontians during the Early Cretaceous, and continued occupation of the shared areas of ecomorphospace by hadrosauroids and (immature) hadrosaurids in the Late Cretaceous suggests that these groups may have occupied similar dietary niches. If conditions necessary for competitive exclusion to occur can be established, then, unlike the original hypothesis of Lucas & Hunt (1989), the decline in sauropod diversity would have been due to competition with both iguanodontians and hadrosauroids. However, it is also possible for resource partitioning to have occurred along other niche dimensions not included in the reconstructed ecomorphospace presented here (e.g., feeding height) because the non-diplodocid sauropods, which showed the greatest overlap with iguanodontians in ecomorphospace, continue to co-occur with iguanodontians in Early Cretaceous deposits (Wedel, Cifelli & Sanders, 2000; D’Emic & Foreman, 2012; Britt et al., 2017). Competitive exclusion or niche partitioning in response to competitive interactions usually occurs on short geological time scales (∼40 Myr) (Benton, 1996), although some studies investigating ecological interactions in the fossil record have been able to produce reasonably compelling cases for competition with good fossil data (Janis, Gordon & Illius, 1994; Tyler & Leighton, 2011; Mallon, 2019). However, North American non-diplodocid sauropods and iguanodontians co-occur on a time scale much greater than would be expected under conditions of ecological antagonism—even when these other examples of possible competition in the fossil record are considered. Even more interesting is that iguanodontians undergo an evolutionary radiation during this time. This is contrary to what would be expected given the overlap observed in reconstructed ecomorphospace and the broad niches of iguanodontians which would implicate competitive interactions between these groups. The similarity in the positioning of non-North American and North American iguanodontians relative to non-diplodocid sauropods suggests similar ecological occupation of these clades across space and further suggests that factors other than ecological antagonism were responsible for the ‘sauropod decline’ in North America. The continued persistence of non-diplodocid sauropods from the Late Jurassic through to the end of the Cretaceous on other landmasses would imply that the overlap observed along PC axes between these clades is more an artefact of the variables selected for the analyses rather than true ecological interactions between these clades. That is, iguanodontians and non-diplodocid sauropods would have experienced ecological niche partitioning along some other niche dimension (e.g., differences in feeding height, environmental preference, reproductive history, migration patterns) not captured by the current analysis.

Interestingly, diplodocid sauropods, which disappear from Early Cretaceous deposits, show less overlap with iguanodontians than non-diplodocid sauropods. Given the absence of diplodocids in Early Cretaceous deposits, under conditions of competitive exclusion it would be expected that this overlap would be more pronounced. The disappearance of diplodocids from North America can more likely be explained by other factors (e.g., climate change), changes in competitive interactions with growth (Woodruff et al., 2018), or a combination of both. The subsequent occupation of previously-sauropod occupied ecospace by immature Late Cretaceous hadrosaurids may provide support for the occurrence of competitive exclusion at immature stages. Such an explanation would require future investigations considering not only adult-adult interactions but also juvenile-juvenile and juvenile-adult interactions once more juvenile sauropod cranial material has been recovered.

Another possibility is that iguanodontians and sauropods did compete, and that this competition was enough to limit sauropods to certain niches or geographic locations but not enough for complete extirpation to occur. The absence of sauropod material from many Late Cretaceous formations from north/north-western North America would support this—currently the only known sauropod skeletal material from the Late Cretaceous is from the Javelina and Black Peaks formations in Texas, Kirtland and Ojo Alamo Formations in New Mexico, the Adobe and Turney Ranch formations in Arizona, and Upper Aguja-lower Javelina equivalent strata in Mexico (McCord, 1997; Montellano-Ballesteros, 2003; Fowler & Sullivan, 2011; D’Emic, Wilson & Williamson, 2011; D’Emic, Foreman & Jud, 2016; Fronimos & Lehman, 2014). Yet, sauropods have not been recovered from Late Cretaceous deposits from Montana or Alberta. The absence of sauropods in Montana is particularly interesting given that during the Late Jurassic sauropods are diverse and abundant (Foster, 2003) compared to ornithischians. The absence of sauropods from these deposits is unlikely a result of sampling biases, at least in some of the well-sampled formations such as the Hell Creek (Montana) and the Dinosaur Park (Alberta). This pattern would more likely reflect differences in environmental preference between clades rather than preservational biases as numerous non-sauropod remains have been recovered from these more interior formations. Potentially, hadrosaurids, although able to occupy environments in which sauropods (specifically non-diplodocid sauropods) are found, such environments are not ideal and limit the amount of competitive pressure hadrosaurids/iguanodontians impose upon sauropods.

It is often proposed that the success of the iguanodontians was due to the possession of complex grinding surfaces based on a series of tooth families, which in derived hadrosaurids each comprise three or four functional teeth in adults and one to three in juveniles and subadults (Erickson & Zelenitsky, 2014; Wyenberg-Henzler, Patterson & Mallon, 2022). Such a complex grinding surface is unlikely to have played any role in early competition between sauropods and iguanodontians as more basal iguanodontians (e.g., Eolambia, Tenontosaurus) only possess one functional tooth per tooth family (McDonald, Wolfe & Kirkland, 2010; McDonald et al., 2017; Thomas, 2015; Prieto-Márquez, Erickson & Ebersole, 2016). However, it is possible that the ability to orally process food was itself sufficient for iguanodontians to take their place among the dominant megaherbivores in Late Cretaceous ecosystems, as sauropods are commonly thought to have relied on gut fermentation and/or gastroliths to process food (Christian, 2002; Wings, 2007; Wings & Sander, 2007; Hummel et al., 2008). Such a proposal would need further in-depth study and the relative importance of processing ability would be largely dependent upon whether or not competitive interactions between these groups could be confidently established.

Another potentially ecologically important difference between iguanodontian and sauropod skulls pertains to their biomechanical capabilities. Separation observed between iguanodontians and diplodocids on PC 4, and variable loadings on this PC axis indicate that distal tooth row lengths (the distance between the distal end of the tooth row and middle of the quadrate) are shorter in iguanodontians. This feature would have increased the biomechanical advantage of the jaw musculature, permitting iguanodontians to consume more mechanically resistant plant materials than diplodocids, which is in-line with the results of previous studies (Ostrom, 1961; Ostrom, 1964; Greaves, 1974; Young et al., 2012; Mallon & Anderson, 2013; Button, Rayfield & Barrett, 2014; Weishampel & Norman, 1989). The greater bite forces produced by iguanodontians would have permitted consumption and processing of tough and soft vegetation. This in turn would indicate a wider niche breadth (as implicated by wide ecomorphospace occupation) for iguanodontians compared to sauropods (especially diplodocids) which may have been restricted to mechanically less resistant plant materials. Whether or not this actually played a role in the decline of sauropods as a result of competitive exclusion or even simply a reduction in geographic range as a result of competition, requires further investigation.

Body size and ecospace

The lack of a clear distinct pattern in species-richness body size distributions between sauropod- and non-sauropod- bearing formations suggests that the presence/absence of sauropods does not dramatically alter the shape of the distribution. Thus, sauropods may have merely been “filling in” the large-bodied end of the size spectrum in iguanodontian-bearing communities. While sauropods may have influenced the distributions of Late Jurassic formations (e.g., Morrison), iguanodontians (and to some extent ceratopsids) filled most of these larger size (>1,000 kg) niches during the Late Cretaceous—at least in North America. This replacement of sauropods with large-bodied ornithischians is also potentially reflected in feeding heights of these taxa. Although it is difficult to more precisely define the nature of sauropod feeding envelopes due to uncertainties surrounding neck posture, all reconstructions agree that sauropods would have fed at heights above ∼3 m (Foster, 2003; Mallon et al., 2013; Barrett, 2014; Stevens & Parrish, 1999; Stevens & Parrish, 2005; Woodruff, 2017). Iguanodontians, particularly the Late Cretaceous hadrosaurids, are reconstructed as feeding between 0 and 4 m. Thus, both sauropods and iguanodontians would have both filled higher-browsing niches within their respective ecosystems. From an ecomorphological perspective, sauropods and iguanodontians also appear to show niche overlap. In combination, these similarities in cranial ecomorphology, feeding height (at least in a broad sense) and body size category, would suggest that iguanodontians were potentially capable of opportunistically filling sauropod niches. However, such an assertion would require further testing involving other lines of evidence such as dinosaur feeding heights and consideration of plant heights through time.

Caveats and future research

Caution must be applied to the interpretation of data points (representing taxa collected from several geological stages), especially when some of those stages lack more than a few data points for a given clade (e.g., Early Cretaceous non-diplodocid sauropods). With the discovery of more material from before the Late Cretaceous of North America, it will be important to conduct follow-up ecomorphological studies to assess how ecospace distribution of the more poorly sampled clades changes with increased sampling. These future studies will also help to further assess whether competitive replacement or opportunistic niche invasion was more likely. For instance, the use of only one ankylosaur specimen from the Late Jurassic is also problematic, as it is impossible to truly assess how much overlap between ankylosaurs and stegosaurs was actually present prior to the extinction of stegosaurs in North America.

Another consideration is that the niches described here are constructed based on multiple taxa from each clade. This means the niche breadth of an individual taxon within a certain clade was not assessed. Thus, species-level interactions were not characterized and niche occupation may have differed between taxon within the same clade. However, from the relative stability of Late Cretaceous ecosystems as commented on by Mallon (2019), it would seem plausible that species from the same clade would occupy relatively similar niches to one another. This said, ecomorphological investigations into niche occupation of different species within a given clade with multiple specimens per taxon would need to be conducted for there to be any certainty in the accuracy of this claim. Such an analysis is presently not possible for all the taxa considered here as some are only represented by a single specimen available for study.

Conflict between the interpretations based on the results of NPMANOVAs (that there is no overlap) and interpretations based on observed overlap within ecomorphospace is another potential problem. This disagreement may relate to the nature of the NPMANOVA which compares the centroids of each group rather than the overall distributions themselves (Hammer & Harper, 2006). As mentioned earlier, this disagreement may also simply reflect differences in morphology that did not result in differences in ecological niche between groups. Dental microwear analyses comparing ankylosaurs and stegosaurs, and sauropods and iguanodontians, may prove a fruitful line of inquiry in the future, as the relative frequencies of pits and scratches on teeth have been shown to reflect differences in feeding style and the mechanical resistance of consumed plant materials (Fiorillo, 1998; Semprebon et al., 2004; Semprebon, Sise & Coombs, 2010; Rivals & Semprebon, 2011; Whitlock, 2011; Mallon & Anderson, 2014). Similar proportions of microwear features on the teeth of stegosaurs/ankylosaurs and sauropods/iguanodontians of the same time interval would further support competitive replacement as a potential explanation for megaherbivore turnover in North America between the Late Jurassic and Late Cretaceous especially if microwear can be sampled and compared between co-occurring taxa from these groups. Stable isotope analyses may also provide useful insights into dietary and habitat preferences of co-occurring taxa and have been used to make dietary inferences in other studies (e.g., Tütken et al., 2004; Fricke & Pearson, 2008; Cullen et al., 2020; Frederickson, Engel & Cifelli, 2020).

As mentioned previously, there are several requirements in addition to similarities in dietary niches that must also be met for competitive exclusion to have been possible (see Methods). This includes resource limitation, overlap between taxa in time (e.g., geological time, seasonality, diel activity), overlap in space, and overlap in feeding envelopes (both between adults and across different ontogenetic stages). Establishing overlap in feeding envelopes may prove especially challenging for some clades. In sauropods, disagreement over neck posture cause result in maximum feeding height estimates differing as much as 4.4 m (Camarasaurus; Foster, 2003; Barrett, 2014; Stevens & Parrish, 1999; Stevens & Parrish, 2005; Woodruff, 2017). Stegosaur feeding envelopes may have extended up to 3.3 m above ground level, 2.3 m greater than the estimate put forth by Foster (2003), as Mallison (2010) has suggested a tripodal feeding posture was feasible based on mechanical computer modelling in Kentrosaurus. Although more definitive feeding height estimates are needed for these clades, should future research reinforce or significantly increase differences in feeding envelopes between the taxa assumed to have overlapping feeding envelops in the present study, this would provide a potential niche dimension for resource partitioning to have occurred along. Future work focusing on ecological interactions during the ‘sauropod decline’ should focus on assessing resource availability and investigating ecological interactions at narrower geographic and temporal scales in addition to considering the impacts of ontogeny on ecological interactions.

It is also important to investigate the potential for other factors as contributors and/or causes behind the ‘sauropod decline’. Even though establishing environmental and/or floral changes as potential factors can be difficult, it is not entirely impossible. Although some research has been conducted on these alternative explanations, as we continue to sample sediments from the Early Cretaceous and the early Late Cretaceous, follow-up assessments of these hypotheses will be necessary. Insight into potential driver(s) behind the ‘sauropod decline’ may also provide information into the factor(s) behind turnover observed in other dinosaur groups, such as large carnivorous theropods (Nesbitt et al., 2019; Holtz, 2021). Similarities in the positioning of clades for analyses with and without non-North American representatives despite differences in community composition between North American ecosystems and those on other landmasses further suggests that non-competitive factors were more important in the North American ‘sauropod decline’. Assessing this will require consideration of other niche dimensions such as feeding height and environmental tolerances to investigate if partitioning is occurring along other niche dimensions.

Conclusions

Megaherbivore composition in the Late Jurassic and Late Cretaceous of North America was very different and the potential differences in ecomorphospace occupation of the major megaherbivore clades from these time intervals had been previously uninvestigated. Competitive replacement had also been previously dismissed as a potential explanation for the Cenomanian and Campanian ‘sauropod decline’ in North America on the basis of morphological differences between sauropods and iguanodontians (Lucas & Hunt, 1989; Barrett, 2014 and references therein). However, general morphological differences are not enough to infer absence of competition between taxa. A lack of competitive potential should be established by comparing traits reflecting ecological niche applicable across a wide taxonomic range. The plausibility of competitive replacement as an explanation for the extinction of North American stegosaurs was also previously unconsidered. This study aimed to investigate ecomorphospace occupation for major megaherbivore clades from the Late Jurassic through to the Late Cretaceous of North America and assess if morphological dissimilarity was too great for competition to have potentially occurred between sauropods and iguanodontians, and stegosaurs and ankylosaurs. Distribution within reconstructed ecomorphospace suggest iguanodontians and sauropods both fed upon tough vegetation at least 1 m above ground level despite differences in processing abilities and the wider niche breadths of the former. Ankylosaurs and stegosaurs also overlapped in dietary niche, feeding on soft-vegetation growing at or below 1 m whereas Late Cretaceous ceratopsids occupied their own distinct dietary niches feeding upon tough vegetation growing at heights below 1 m. These results suggest that morphological dissimilarity between taxa, particularly between iguanodontians and sauropods, was not enough to prevent competition from occurring between these clades, as had originally been claimed by Lucas & Hunt (1989) when dismissing the competitive replacement hypothesis as an explanation for the ‘sauropod decline’ in North America. Whether this overlap is reflective of competitive replacement, opportunistic exploitation of recently vacated niche space, or niche partitioning along unaccounted niche dimensions is difficult to determine without further investigation. However, the absence of any clear differences in the shape of body-size species richness distributions between sauropod-bearing and non-sauropod bearing ecosystems combined with (1) the ability of both sauropods and igaunodontians to feed above 3 m and (2) relatively little change in ecospace relationships with the addition of non-North American taxa, suggest that Late Cretaceous iguanodontians may have opportunistically filled some or all of the higher-browsing, large-bodied niches previously filled by sauropods in Late Jurassic ecosystems. However, such a claim will require further testing to assess exactly how such niches may have been vacated—either a result of ecological competition, an environmental change, floral change or some other factor.

Future research will be needed to more fully investigate competitive potential and associated underlying assumptions necessary for competition to occur (e.g., resource limitation) with larger sample sizes, especially for all groups prior to the Late Cretaceous, and consideration of other niche dimensions, in order to assess which of these explanations was more likely. Investigations into reconstructed feeding heights of these clades, particularly those of sauropods, will also be important for establishing competitive potential as differences in feeding height may have reduced competitive pressures placed on higher-browsing non-diplodocids by basal iguanodontians. Interactions between clades at different ontogenetic stages will also need consideration as competitive interactions may have changed between clades with maturity. Differences in other niche dimensions, such as environmental preferences, should also be considered. It is also possible for other factors to have caused or contributed to the ‘sauropod decline’ such as floral or climactic changes should also be considered. In short, distribution within reconstructed ecomorphospace suggests morphological dissimilarity likely did not preclude dietary overlap between coeval representatives of major Late Jurassic and major Late Cretaceous megaherbivore clades as Lucas & Hunt (1989) had previously asserted and so further research into the competitive replacement hypothesis and other hypotheses is needed to improve our understanding of the events behind the megaherbivore turnover observed between the Late Jurassic and Late Cretaceous in North America.

Supplemental Information

File S1 Complete list of institutional abbreviations and referenced summary of R packages used in analyses

Click here for additional data file.

Supplemental Information 2 Measurement description, raw measurements and R code used in analyses

Click here for additional data file.

Supplemental Information 3 Excel workbook containing spreadsheets with degrees of freedom, test statistics and p-values for each iteration of NPMANOVAs conducted

Click here for additional data file.

Figure S1 Principal component plot for PC1 versus PC4 representing reconstructed ecomorphospace during the Late Cretaceous (A), Early Cretaceous (B), and Late Jurassic (C) for North American taxa

Convex hulls are drawn around specimens from major clades across all time periods. Image credits after Fig. 2. Measurement descriptions and raw data are available in Fig. 1 and Tables S1 and S2.

Click here for additional data file.

Figure S2 Loadings plots for principal components axes of North American dataset

Measurement abbreviations as in Fig. 1 and Table S1.

Click here for additional data file.

Figure S3 Principal component plot for PC1 versus PC4 representing reconstructed ecomorphospace during the Late Cretaceous (A), Early Cretaceous (B), and Late Jurassic (C) for North American and non-North American taxa

Convex hulls are drawn around specimens from major clades across all time periods. Abbreviations denoting location of non-North American taxa: A, Argentina; B, Brazil; C, China; M, Mongolia; S, Spain. Image credits after Fig. 2. Measurement descriptions and raw data are available in Fig. 1 and Tables S1, S2 and S3.

Click here for additional data file.

Figure S4 Loadings plots for principal components axes of analysis of North American and non-North-American taxa

Click here for additional data file.

Constructive criticism of early drafts of this paper were provided by L. Leighton, C. Sullivan and two anonymous reviewers. My thanks also go to A. Farke, T. Holtz and N. Longrich for their constructive reviews of the most recent manuscript as well as M. Meegaskumbura for their editorial assistance. Helpful conversations and methodological support were offered by B. Arnaout, N. Campione, D. Fraser, L. Leighton and J. Mallon. Thanks to curators and their assistants at AMNH, CMN, GPDM, MOR, ROM, TMP, USNM, and UALVP during the collection of the ornithischian dataset during my Masters research. Special thanks to A. Knapp for access to materials and models and J. Mallon for access to ornithischian snout photos.

Variable Abbreviations

as-mq anterior snout to middle quadrate distance

cp-jj coronoid process to jaw joint distance

cs-mq cropping surface to middle quadrate distance

mt-mq mesial tooth row to middle quadrate distance

dh dentary height

dt-mq distal tooth row to middle quadrate distance

oh occiput height

ppb paroccipital process breadth

qb quadrate breadth

sk skull height

sw snout width

SSI snout shape index (after Mallon & Anderson, 2013; Wyenberg-Henzler, 2020)

Institutional Abbreviations

AMNH American Museum of Natural History, New York, New York;

CMN Canadian Museum of Nature, Ottawa, Ontario;

GPDM Great Plains Dinosaur Museum, Malta, Montana;

MOR Museum of the Rockies, Bozeman, Montana;

ROM Royal Ontario Museum, Toronto, Ontario;

TMP Royal Tyrrell Museum of Palaeontology, Drumheller, Alberta;

UALVP University of Alberta, Edmonton, Alberta;

USNM Smithsonian Institution, National Museum of Natural History, Washington, D.C.

Additional Information and Declarations

Competing Interests

Author Contributions

Data Availability

The author declares there are no competing interests.

Taia Wyenberg-Henzler conceived and designed the experiments, performed the experiments, analyzed the data, prepared figures and/or tables, authored or reviewed drafts of the paper, and approved the final draft.

The following information was supplied regarding data availability:

The code used in all analyses and the raw measurements are available in the Supplementary Tables.

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
