# Peer review of "Ecomorphospace occupation of large herbivorous dinosaurs from Late Jurassic through to Late Cretaceous time in North America"

_PeerJ, doi:10.7717/peerj.13174_

## Round 0.1 · original submission · Minor Revisions

I have received three reviews for your paper and they all agree that your study is worthy of publication, albeit with some revisions. The comments of all three reviewers are encouraging, constructive and reasonable and hence please address these as much as possible. The third reviewer’s point that “not all variation is functionally significant” is quite apt, and I hope you will consider the diet related analyses that is proposed in addition to what you have already done. This will also help indirectly address the point the first reviewer raises “competition focuses primarily on diet, which is of course quite important. But...competitive exclusion can also be mediated by things such as reproductive traits, or other life history traits (e.g., growth rate), or resistance to predation”.

·

Basic reporting

This manuscript by Wyenberg-Henzler, titled "Ecomorphospace occupation of large herbivorous dinosaurs from Late Jurassic through to Late Cretaceous time in North America," provides an interesting and informative approach to discussions of ecological competition and replacement in major dinosaur groups. The paper is clearly written, with complete citations of relevant papers, and an appropriately conservative interpretation of the results.

Experimental design

The methods presented here are out of my area of immediate expertise, so I cannot provide substantive comment in this area.
The research question is well defined and appropriately tied to previous discussions in the field.
The methods are sufficiently described, and appropriate data are available.
The paper mentions concerns about sample size for some clades. For time periods and regions when a particular species had low diversity, there's probably not much that can be done about it. In other cases, would it be possible and appropriate to use related taxa from other times or regions as a proxy? Early Cretaceous sauropods are one particular example -- there is only one in the sample (Abydosaurus), but there are of course other taxa known from postcrania. Could skulls of related taxa (e.g., from Europe) be used as substitutes? This is not ideal, of course, but might be helpful for the particularly important segment of the Early Cretaceous.

Validity of the findings

All required data are provided, and seem to be sound.
Conclusions are appropriately constrained given the data and results.

Additional comments

This is briefly mentioned, but it would be worth further emphasizing that there can be avoidance of competition by inhabiting different environments or cycling through landscapes at different times of year.

The issue of competition focuses primarily on diet, which is of course quite important. But...competitive exclusion can also be mediated by things such as reproductive traits, or other life history traits (e.g., growth rate), or resistance to predation.... This is mentioned in part in the discussion, but I think it is important to focus on how other traits might displace sauropods other than diet. We don't know a lot about this of course for most dinosaurs, but it could be yet another explanatory factor.

Overall, any suggestions I have are pretty minor. The overall work is very nicely done!

·

Basic reporting

An excellent quantitative analysis to examine the hypothesis of competitive evolution in the history of North American large-bodied herbivorous dinosaurs.

The text is clear and well-written. The material and references are up-to-date. The flow of the presentation of the analyses and their interpretation is reasonable and sensible.

Experimental design

The measurements chosen for study are reasonable (and previously used in similar such analyses); the statistics used are appropriate for the type of data; and the results are fairly interpreted. The figures succinctly summarize the data well. Of special note: the author is very clear on the current limitations of the data and the necessity to examine these on a broad scale for that reason.

Validity of the findings

The conclusions are sound given the experimental results. The data and methods are described in sufficient detail to allow an independent researcher to duplicate this study.

Additional comments

Many apologies for the lateness of this review.

If the author continues this work into the future, it would be interesting (if sufficient data density allowed) to partition the Late Cretaceous into earlier (Cenomanian-Turonian) and later (Campanian-Maastrichtian) divisions. This would reflect the time before the rise of megaherbivorous ceratopsians versus the age of the Ceratopsidae. There is a similar transition in the makeup of the apex predators at these same divisions (Nesbitt et al. 2019 in the manuscript references; also, Holtz 2021).

Holtz, T.R., Jr. 2021. Theropod guild structure and the tyrannosaurid niche assimilation hypothesis: implications for predatory dinosaur macroecology and ontogeny in later Late Cretaceous Asiamerica. Canadian Journal of Earth Sciences 58: 778-795. doi: 10.1139/cjes-2020-017

COMMENTS, CORRECTIONS, SUGGESTIONS
p.7, line 44 Although not universally followed, the preference is to use Ma only for specific dates (in the context of “millions of years ago”) and Myr for durations. See, for instance, Aubry et al. 2009:
Aubry, M.-P., J.A. Van Couvering, N. Christie-Blick, E. Landling, B.R. Pratt, D.E. Owen & I. Ferrusquía-Villafranca. 2009. Terminology of geological time: establishment of a community standard. Stratigraphy 6: 100-105. https://www.ldeo.columbia.edu/~ncb/Selected_Articles_all_files/25_Stratigraphy.6.100.pdf
p.7, line 59 As useful as it would be (especially for terrestrial paleontologists), there is no formal epoch “Middle Cretaceous”. As such, the “middle” here should be lowercase (and indeed, “mid-Cretaceous” might be preferred).
p.8, line 63 A very important point, especially as the skeletal, dental, and footprint records all fail to find stegosaurs and non-titanosauriform sauropods in Early Cretaceous North America.
p. 13, line 190 These “blocks of time” are simply the standard Epochs of international stratigraphy. You should note, however, that you are excluding the Berriasian through the Hauterivian in your Early Cretaceous. (Also, the base of the Barremian is currently regarded as ~129.4 Ma: https://stratigraphy.org/gssps/)
p. 14, line 207 While stegosaurs are often regarded as low-browsers (and almost certainly were primarily such), Mallison (2010) has shown that a tripodal feeding posture in this clade is at least feasible. It might be interesting to see what your results yield if you include the possibility of mid-level feeding in Stegosauria.
Mallison, H. 2010. CAD assessment of the posture and range of motion in Kentrosaurus aethiopicus Hennig 1915. Swiss Journal of Geosciences 103: 211-233. Doi: 10.1007/s00015-010-0024-2
p. 21, line 371 It is worth noting that although you have only one Jurassic ankylosaur in your analysis (Gargoyleosaurus), at least one more is known (Mymoorapelta) which couldn’t be included in your metrics as there is nothing yet like a complete skull known for it. Thus, lack of overlap might be less if we knew a bit more about Morrison ankylosaur cranial diversity.
p. 22, line 386 Did Benton really claim that competitive interactions only occurred over 1-10 years? I cannot find this statement in his text. I believe this might be a typo, and you meant ~100-101 Myr.

·

Basic reporting

OK

Experimental design

OK

Validity of the findings

OK

Additional comments

Review, “Ecomorphospace occupation of large herbivorous dinosaurs from Late Jurassic through to Late Cretaceous time in North America” by Taia Wyenberg-Henzler

“competitive replacement of sauropods by hadrosauroids”

I’m happy to recommend this paper for publication but have a few suggestions for the author, made on a take-it-or-leave-it basis. If you think they’re good ideas, feel free, if not, do as you like.
My first instinct in reading this paper is that “oh of course it can’t be competitive displacement.” The diplodocoids and stegosaurs are diverse in the late Jurassic and completely disappear in the earliest Cretaceous, which strongly implies a major extinction near the J-K boundary and opportunistic replacement by other clades. I may be biased by studying mass extinction, but in my mind the record here reminds me of Paleocene mammals and dinosaurs- we don’t even need to discuss whether there’s competitive displacement, because there’s no evidence these taxa ever met. I guess my take is that displacement/opportunistic replacement is best tested by looking at the temporal distribution of the taxa, rather than morphology- no temporal overlap, no competition.
Anyway, maybe I’m wrong, but in that light, I think it might be better to focus the study slightly differently? Assuming there’s opportunistic replacement rather than competitive displacement, how do megaherbivores change over time: how do they compare in terms of morphospace overlap, but also overall morphospace occupation? Are these communities functioning similarly or differently from the Jurassic to the Cretaceous?
My sense of the data is that
(1) there’s a distinct shift, which is likely due to both (i) different ways of solving the same or similar problems (e.g. sauropod gastric mills versus hadrosaur dental batteries) but also (ii) the groups solving different problems (lots of huge high feeders versus lots of smaller low feeders);
(2) Not only is there a shift in what the dinosaurs are doing, the Cretaceous ones just do a wider range of stuff.
Anyway, in my mind it might make more sense to focus more on these angles.
With respect to the analyses. I’m not a primarily a specialist in morphometrics, which limits my ability to comment in great detail on the methods; others might be able to comment more usefully. Overall it looks fine, but a couple of thoughts.
Morphometrics is tricky in part because not all variation is necessarily functionally significant. For example, the casque of an oviraptorid or a Parasaurolophus will increase the skull disparity of their clades but these things are probably signaling about sexual selection, rather than feeding.
So one approach is to focus on a couple variables of known functional significance. Friedman’s rather elegant paper on fish extinction in PNAS does this- rather than focusing on lots of different variables he takes two- overall size (using mandible height*width as a proxy for size) and mechanical advantage (ratio of mandible length to the distance between the cotyle and the coronoid process). You’ve got that data so it might be fairly easy to use that approach?
My point is not to say “you’re doing it wrong” but it can be informative to measure things in different ways. If different approaches to measuring something give you broadly similar results, you’re probably onto something- rather than quibbling about the best way to analyze data (which statisticians can and will do forever) your results can be considered more robust if measuring things in different ways yields broadly congruent results. For example, if you measured the birds of the Amazon either in terms of number of species, number of families, skull disparity, size disparity, dietary preferences, or even range of colors, they’re going to be far more diverse than those of Alberta. So we can feel very confident there’s really high bird diversity there, it’s not an artifact resulting from the choice of what we measure.
Another easy approach to examine some of these problems might just be to compare overall size of these animals? The Benson et al. data would give you either mass and/or femur length and you could just plot the mass distributions of these creatures? Body mass is an interesting thing to study because it is correlated with so many differerent variables- locomotion, lifespan, reproductive strategy. Big animals are different from small animals in lots of ways, so when you see a wide range of sizes represented (or not) that tells you something about overall diversity. Similarly, overlap or lack thereof may tell you something. This would be quick-and-easy but I think could nicely complement the skull data and make the paper stronger.
NL

---

## Round 0.2 · accepted · Accept

The manuscript now meets all the editorial requirements and I am happy to accept your excellent paper. It is not often authors are so willing to perform suggested analyses to such a high standard. I wish you the best with your work on dinosaurian paleoecological studies.

·

Basic reporting

The author has sufficiently addressed my comments on the first draft.

Experimental design

The author has sufficiently addressed my comments on the first draft.

Validity of the findings

The author has sufficiently addressed my comments on the first draft.

Additional comments

The author has sufficiently addressed my comments on the first draft.

·

Basic reporting

The revised manuscript improves upon what I had already regarded as a quite good study. The language is clear and professional. The graphics and tables include all the information necessary to evaluate the statistical analyses conducted. And while the paper represents just a part of the author's ongoing research in dinosaur paleoecology, it is self-contained with a specific set of hypotheses to test, and a presentation and interpretation of said tests (and their limitations).

Experimental design

The author is explicit in their choice of measurements to study, and methods to use (or in some cases, not to use: for instance, they make clear the reasons for their decision not to use pPCA-style analyses.)

The hypotheses being examined are clearly stated, and the conditions under which they would be supported or rejected specified.

Validity of the findings

The conclusions stem reasonably from the statistics and plots. Equally important, the author explicitly states the limitations of the data and analyses, and areas in which future studies might clarify issues which are still ambiguous.

Additional comments

It is my opinion that this research will represent a significant foundation upon which future studies of dinosaurian paleoecological studies might be built.